# TriSAM: Tri-Plane SAM for zero-shot cortical blood vessel segmentation in VEM images

## Abstract

In this paper, we address a significant gap in the field of neuroimaging by introducing the largest-to-date public benchmark, BvEM, designed specifically for cortical blood vessel segmentation in Volume Electron Microscopy (VEM) images. The intricate relationship between cerebral blood vessels and neural function underscores the vital role of vascular analysis in understanding brain health. While imaging techniques at macro and mesoscales have garnered substantial attention and resources, the microscale VEM imaging, capable of revealing intricate vascular details, has lacked the necessary benchmarking infrastructure. As researchers delve deeper into the microscale intricacies of cerebral vasculature, our BvEM benchmark represents a critical step toward unraveling the mysteries of neurovascular coupling and its impact on brain function and pathology. The BvEM dataset is based on VEM image volumes from three mammal species: adult mouse, macaque, and human. We standardized the resolution, addressed imaging variations, and meticulously annotated blood vessels through semi-automatic, manual, and quality control processes, ensuring high-quality 3D segmentation. Furthermore, we developed a zero-shot cortical blood vessel segmentation method named TriSAM, which leverages the powerful segmentation model SAM for 3D segmentation. To lift SAM from 2D segmentation to 3D volume segmentation, TriSAM employs a multi-seed tracking framework, leveraging the reliability of certain image planes for tracking while using others to identify potential turning points. This approach, consisting of Tri-Plane selection, SAM-based tracking, and recursive redirection, effectively achieves long-term 3D blood vessel segmentation without model training or fine-tuning. Experimental results show that TriSAM achieved superior performances on the BvEM benchmark across three species.

## 1 Introduction

With around 2% of body weight, our brain receives around 20% of blood supply. With inadequate blood supply, such as stroke, an average patient loses ∼1.9 million neurons (Saver, 2006). Alterations of blood vessel structures are observed in many brain diseases, such as Alzheimer's and vascular dementia (Kalaria, 2010). Thus, blood vessels in the brain have been extensively investigated by a variety of imaging methods with different resolutions and structural details (Figure 1a). Compared to the macro-level imaging (*e.g.*, CT and MRI (Dyer et al., 2017; McDonald & Choyke, 2003)) and mesoscale-level imaging (*e.g.*light microscopy Lochhead et al. (2023)), Volume Electron Microscopy (VEM) (Peddie & Collinson, 2014; Peddie et al., 2022) can reveal the detailed ultrastructure including endothelial cells, glial cells, and neurons (Figure 1b).

Traditionally, the imaging methods at the macro and mesoscale are widely used and have produced a large amount of data, and a variety of image segmentation algorithms, public datasets, and evaluation methods have been developed  (Goni et al., 2022; Moccia et al., 2018). Recently, owing to the rapid improvement of imaging technology, the sample size of VEM is significantly increased covering all the layers of the cerebral cortex of mouse (The MICrONS Consortium et al., 2023)and human brain (Shapson-Coe et al., 2021), as well as the whole brain of fly (Dorkenwald et al., 2023). However, to robustly analyze blood vessels in VEM images, existing segmentation methods suffer from two major challenges: the diversity of the image appearance due to variations in the involved imaging pipeline including sample preparation, and the complexity of blood vessel morphology.

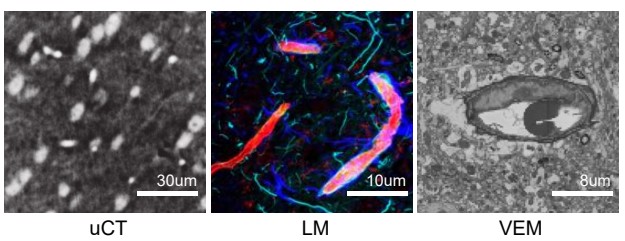 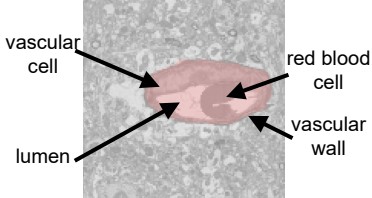

(a) Blood vessel imaging with different modalities  (b) Detailed structures in VEM

Figure 1: Comparison of imaging modalities for blood vessel analysis. (a) Both microtomography (μCT) Dyer et al. (2017) and light microscopy (LM) Lochhead et al. (2023) can capture sub-micron resolution of blood vessels in the cortex, but are unable to provide the ultrastructure details. (b) Volume electron microscopy (VEM) can achieve a higher resolution to show unbiased details of the vasculature including all the cell types and their details are visible for further investigations.

To accelerate the method development, we first curate the BvEM dataset, the largest-to-date public benchmark dataset for cortical blood vessel segmentation in VEM images. The VEM image volumes of the BvEM dataset are from recent publications, which are the largest for each of the three mammal species: mouse, macaque, and human acquired at different VEM facilities. We downsampled the volumes to a consistent resolution and performed extensive blood vessel annotation, including manual proofreading, semi-automatic segmentation error correction, and quality control, involving multiple rounds of scrutiny by neuroscience experts to ensure accuracy and completeness.

We then propose a zero-shot 3D segmentation method named TriSAM based on the Segment Anything Model (SAM) (Kirillov et al., 2023) which is able to segment objects in the image given points or a bounding box as the input prompt. With a multi-seed tracking framework, our proposed TriSAM consists of three components: Tri-Plane selection, SAM-based tracking, and recursive redirection. Given the fact that tracking along blood vessel flow direction is easier, our Tri-Plane selection method chooses the best plane for tacking. To further exploit the 3D blood vessel structure and ensure long-term tracking, we propose to change the tracking direction at the potential turning points with a recursive redirection strategy. The proposed method achieves state-of-the-art performance compared to the baseline methods on the proposed BvEM benchmark across all three species.

## 2 BvEM Dataset

**Dataset Design.** To foster new methods tackling the challenges, we built our **BvEM dataset** on top of the largest publicly available VEM image volumes for samples from each of three mammal species: visual cortex from an adult mouse The MICrONS Consortium et al. (2023), superior temporal gyrus from an adult macaque Loomba et al. (2022), and temporal lobe from an adult human Shapson-Coe et al. (2021). Each dataset was acquired with different protocols at different VEM facilities and we refer readers to the respective papers for more details.

**Image Volume Standardization.** We first downsampled all three VEM image volumes to a near-isotropic resolution (∼200-300 nm) along each dimension, which is a good balance between rich image details for biological analysis and a manageable dataset size for computation and dissemination for AI researchers. Then, we trimmed off the rows and columns across all slices near the image boundary of the mouse and human data, where the blood vessels are hard to annotate due to the missing image content. We directly used the macaque dataset which was well-trimmed by its authors. As shown in Figure 2, the imaging quality and the appearance of blood vessels vary drastically across these three datasets, showcasing the cutting-edge large-scale VEM pipelines in the field.

**Blood Vessel Annotation.** We annotated the 3D instance segmentation for blood vessels in the processed image volumes above based on available results. (1) *Initial annotation.* The human data paper provides a manual proofread blood vessel segmentation, which has many segments with incomplete shapes and disconnected blood vessel instances due to missing segmentation annotation in between. The mouse and macaque dataset papers only provide dense 3D instance segmentation and we manually picked out the blood vessel segments as the initial annotation. Due to the large scale of the BvEM-Mouse and BvEM-Human datasets, we followed the practice of the human dataset paper

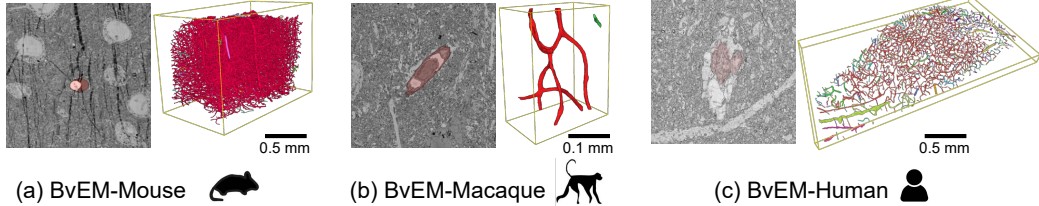

(a) BvEM-Mouse          (b) BvEM-Macaque          (c) BvEM-Human

Figure 2: BvEM Dataset. We compiled the largest publicly available VEM dataset for each of the (a) mouse The MICrONS Consortium et al. (2023), macaque Loomba et al. (2022), and human Shapson-Coe et al. (2021) samples acquired by three different VEM labs. For each dataset, we downsampled the image volume to a similar resolution and trimmed off boundary regions. Then, we corrected the existing ground truth or instance segmentation results for each blood vessel instance (displayed in different colors).

Table 1: Dataset Statistics. Despite the difference in the scale and the geometry of the image volume, the largest blood vessel instance (rendered in red) is significantly bigger than the rest combined.

| Sample | Microscope | Resolution (nm) | Volume size (voxel) | Length: max/sum (mm) |
|--------|-----------|-----------------|---------------------|----------------------|
| Mouse | TEM | $320\times 256\times256$ | $2495\times3571\times2495$ | 1.6/1.7 |
| Macaque | SBEM | $240\times176\times176$ | $450\times1271\times995$ | 713.3/714.5 |
| Human | MultiSEM | $264\times256\times256$ | $661\times7752\times13500$ | 107.2/126.7 |

to only annotate every 4 slices, where the z-dimension resolution is around $1\mu$m. (2) *Semi-automatic segmentation error correction.* We used the VAST lite software Berger et al. (2018) which can accelerate the proofreading process by using provided segmentation results as the drawing or filling template instead of delineating segments manually. (3) *Quality control.* We used 3D visualizations and skeleton analysis to detect remaining segmentation errors. Each dataset was proofread by two neuroscience experts in multiple rounds until no disagreement.

**Dataset Statistics.** We summarize the dataset details in Table 1, where the dimension is in the "zyx" order. The BvEM-Macaque data has around 0.5G voxels, and the BvEM-Mouse and BvEM-Human image data are around 80 and 121 times bigger than it. From the blood vessel instance segmentation annotation, we automatically extracted skeleton centerlines with the Kimimaro software (Silversmith et al., 2021) and computed the length for each blood vessel instance. Due to the hyper-connectivity nature of cortical blood vessels, the length of the largest instance is around 99%, 95%, and 85% for each dataset. Note that the BvEM-Human dataset is a thin slab, where many blood vessels are disconnected due to the limited spread.

## 3 METHOD

Conventional blood vessel segmentation heavily depends on a substantial volume of manually annotated data, a resource that is notably scarce in the existing literature. Therefore, we propose a zero-shot 3D blood vessel segmentation method based on SAM (Kirillov et al., 2023) which achieves superior performance without model training or fine-tuning.

### 3.1 PROBLEM FORMULATION: 3D SEGMENTATION AS MULTI-SEED TRACKING

To apply SAM to 3D VEM images, a naive method is to combine SAM with a tracking algorithm. However, blood vessels are sensitive to split error requiring robust long-term tracking which is a remaining challenge in object tracking literature. Another possible solution is to use a 3D adaptor with 2D SAM features which requires a large amount of labeled data for training. However, labeled training data is limited since the annotation for VEM images needs the involvement of experts. To this end, we formulate 3D segmentation as a multi-seed tracking framework shown in Figure 3a that doesn't require any annotated data or robust long-term tracking algorithms. To achieve long-term tracking with multi-seed tracking, we leverage the fact that one of the planes is reliable for tracking while other planes indicate potential turning points. In particular, different from videos, the

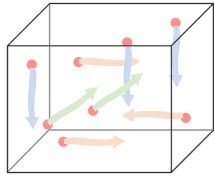 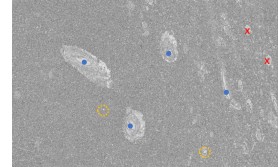 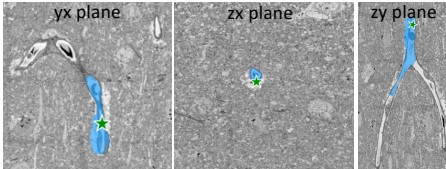

(a) Multi-seed tracking     (b) Initial seed generation     (c) Segmentation in different planes

Figure 3: Problem formulation and challenges. (a) The 3D segmentation task can be formulated as the multi-seed object tracking problem. There are two main challenges for such an approach: (b) generating high-quality initial seeds (blue dots) and minimizing the false positive (red crosses) and false negative errors (yellow circles); (c) adaptively selecting tracking directions as the quality of SAM segmentation results can vary significantly.

VEM data can be interpolated from different planes (e.g. yx/zy/zx planes). Tracking along some planes is extremely difficult since the shape and scale change dramatically. Therefore, it is essential to select which plane should be used for tracking. To this end, we propose a *Tri-Plane Selection* method to select the best plane for tracking based on the potential blood flow direction. To achieve long-term tracking of blood vessels, a *Recursive Redirection* method is proposed to keep tracking at potential turning points. Based on these two components, a simple *SAM-based Tracking* is effective in achieving long-term 3D blood vessel segmentation.

### 3.2 TriSAM 3D Segmentation

The proposed TriSAM 3D segmentation method has three components: Tri-Plane selection, SAM-based tracking, and recursive redirection. The initial seeds are first generated automatically or manually. Then, TriSAM 3D segmentation is performed on each seed and the segmentation results are combined as the final segmentation. The pipeline of the proposed method is shown in Figure 4.

**Initial Seed Generation.** Initial seeds can be effectively generated with global color thresholding since the pixels of blood vessels are brighter than the background. Therefore, pixels darker than a threshold are considered as background while others are seed candidates. To improve efficiency, we only keep the center of each connected component as the final seeds. In practice, the threshold is set to $\eta I_m$ where $I_m$ is the maximum intensity of the volume.

**Tri-Plane Selection.** Rather than videos, 3D VEM images can be interpreted in different planes. In practice, tracking along some of the planes is hard since the shape and scale of the blood vessels change dramatically. As shown in Figure 4, tracking along the bottom plane is hard because the shape is complex and changes quickly. This phenomenon can be further confirmed by the experiment shown in Figure 6 in which tracking along the $y$ axis is more effective than tracking along the $x$ axis. Tracking along the plane according to the blood flow direction is ideal and this can be achieved based on choosing a plane with smaller segmentation as shown in Figure 4. Furthermore, we take into account the probability associated with the SAM segmentation result, as it reflects the confidence level in identifying blood vessels. Overall, the plane with the smallest segments and a probability higher than the threshold $\tau$ is selected as the best plane.

**SAM-based Tracking.** By tracking along the optimally selected plane, changes in shape and size of the blood vessel are more stable compared to other planes. Therefore, we use the enlarged bounding box and the center of the segment as the prompts for the next slice. In particular, we first pad the bounding box of the segment as the bounding box prompt and compute the center of the segment as the point prompt. These prompts are used for segmenting blood vessels in the next slice. The tracking will stop when the probability of the segment is less than a threshold $\tau$.

**Recursive Redirection.** SAM-based tracking is simple and fast for short-term tracking. But it may fail when shape and scale change dramatically as shown in Figure 3c. To enable long-term tracking, we propose a recursive redirection method to exploit the 3D blood vessel structure, which keeps track of potential turning points in blood vessels. Specifically, for each seed, we maintain a turning point tree where the root is the initial seed. In each iteration, we sample a left node in the turning point tree and perform Tri-Plane selection and track along the chosen plane. As blood vessels may change directions within this iteration, we then split this node to obtain new potential turning points. We find that the non-best planes potentially identify turning points or endpoints accurately as shown

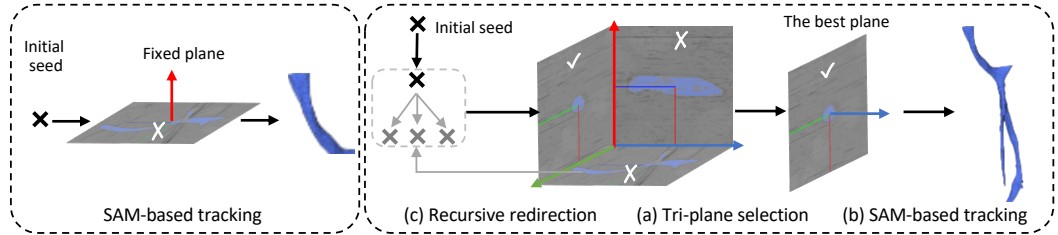

Figure 4: Left: SAM-based tracking. Right: our proposed TriSAM. (a) Tri-Plane selection is first proposed to select the best plane for tracking. (b) SAM-based tracking leverages SAM to perform short-term tracking given a seed location and a tracking direction. (c) Recursive redirection exploits potential turning points for long-term tracking.

in Figure 4, Therefore, we propose to generate the turning points according to the segmentation results at non-best planes. In particular, when tracking along the z-axis, we apply SAM to segment blood vessels in the zx and zy planes. Potential turning points are identified as points in these segments with the smallest and largest z-values.

## 4 EXPERIMENTS

In this section, we present TriSAM's experimental results. We begin with settings and implementation details. Then we show the benchmark results. Finally, we perform ablations for further analysis.

### 4.1 EXPERIMENTAL SETTINGS AND IMPLEMENTATION DETAILS

**Dataset Split.** We used the whole dataset for evaluation. For the supervised setting, we randomly sampled subvolumes are annotated. We sampled 4 subvolumes of size $256 \times 256 \times 256$ for the BvEM-Macaque dataset amounting to around 10% of the dataset, and 12 subvolumes of size $384 \times 384 \times 384$ for each of the BvEM-Mouse and BvEM-Human datasets amounting to around 1% of each dataset. These annotated subvolumes were divided into a 3-1 train-valid split.

**Implementation Details.** Since the volume is too large, non-overlapping subvolumes are used for segmentation and the results are fused to form the final prediction. For BvEM-Macaque, we directly predict the whole volume. For others, $k \times 1024 \times 1024$ subvolumes are used since $1024 \times 1024$ is the default resolution for SAM where $k = 661$ for BvEM-Human and $k = 818$ for BvEM-Mouse. For preprocessing, we utilize defliker method and temporal smoothing on the z-axis to remove the artifacts and missing slices. For postprocessing, holes are filled and small connected components are removed after the segmentation. Global color thresholding is used to generate the initial seeds and $\eta$ is set to 0.98 as shown Figure 7 (left). Given the significantly greater complexity of human blood vessels, we employ the manually generated seeds for the challenging BvEM-Human dataset. For SAM-based tracking, the probability threshold $\tau$ is set to 0.8 according to the experiment shown in Figure 7 (right). Unless explicitly stated otherwise, we use MobileSAM (Zhang et al., 2023) instead of the standard SAM (Kirillov et al., 2023) in all conducted experiments to improve the inference speed. All experiments are conducted on an NVIDIA-A100 GPU.

**Evaluation Metrics.** Following Weigert et al. (2020), we use Precision, Recall, and Accuracy as the metrics to evaluate the performance:

$$Precision = \frac{TP}{TP + FP}, \quad Recall = \frac{TP}{TP + FN}, \quad Accuracy = \frac{TP}{TP + FP + FN}, \quad (1)$$

where $TP$, $FP$, and $FN$ are instance-level true positive, false positive, and false negative respectively. We use instance-level metrics since it is more sensitive to split errors. In particular, the Hungarian algorithm is used to match ground-truth instances and predicted instances with negative Accuracy as the cost matrix.

---

[1]Results from original dataset papers. BvEM-Mouse and BvEM-Macaque are automatically segmented while BvEM-Human is manually labeled.

Table 2: Benchmark results on the proposed BvEM dataset. We evaluate the initial blood vessel annotation to show the amount of proofreading effort. We compare the proposed TriSAM with various baseline methods under fully supervised with limited training data, unsupervised, and zero-shot settings.

| Method | Training Data | BvEM-Mouse | | | BvEM-Macaque | | | BvEM-Human | | |
|---|---|---|---|---|---|---|---|---|---|---|
| | | Pre | Rec | Acc | Pre | Rec | Acc | Pre | Rec | Acc |
| Initial Annotation [1] | Unknown | 93.74 | 36.62 | 35.74 | 1.65 | 23.17 | 1.57 | 100.00 | 25.68 | 25.68 |
| 3D UNet | Limited | 13.45 | 0.91 | 0.67 | 16.04 | 89.65 | 15.75 | 68.63 | 2.46 | 2.43 |
| nnUNET | Limited | 3.54 | 49.55 | 6.59 | 24.34 | 29.57 | 23.74 | 3.44 | 23.20 | 5.86 |
| Color Thresholding | None | **86.45** | 37.32 | 35.26 | **95.14** | 21.65 | 21.42 | **41.77** | 1.92 | 1.87 |
| MAESTER | None | 2.16 | 18.95 | .94 | 22.08 | 40.30 | 16.64 | 0.29 | 5.03 | 0.27 |
| SAM + IoU Tracking | External | 63.59 | 0.27 | 0.27 | 74.39 | 1.89 | 1.88 | 18.19 | 23.58 | 12.92 |
| **TriSAM** | External | 84.12 | **66.75** | **59.28** | 78.41 | **74.97** | **62.14** | 31.35 | **25.57** | **16.39** |

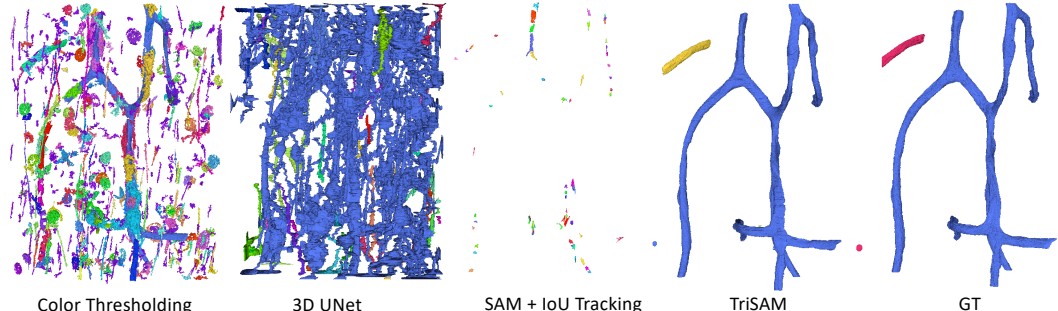

| Color Thresholding | 3D UNet | SAM + IoU Tracking | TriSAM | GT |
|---|---|---|---|---|

Figure 5: Instance segmentation results of comparison methods on BvEM-Macaque. Different colors indicate different instances. Color thresholding and 3D UNet often produce false positives, whereas SAM + IoU tracking tends to miss a significant portion of blood vessels. Among the comparison methods, TriSAM segmentation stands out as the most effective.

## 4.2 BENCHMARK RESULTS

**Methods in Comparison.** We compare the proposed TriSAM with both zero-shot baselines and supervised methods. The compared zero-shot methods are global color thresholding and SAM + IoU tracking. For color thresholding, we first perform (3D) Gaussian blurring with $\sigma = 1$ on 3D chunks (10x512x512). Then we label all voxels that are 3 standard derivation above mean as positive. Finally, connected components with less than 1000 voxels are filtered out. For SAM + IoU Tracking, we segment all objects in each z-slice of the dataset using automatic mask generation. Then we track each blood vessel using the first labelled slice as seeds. Our simple tracking algorithm finds the mask in next slice with maximum IoU with the current slice. If the max IoU is above a threshold, we assign this mask to the current object and continue tracking. We also tried SAM + IoU tracking with microSAM Archit et al. (2023b) weights that have been finetuned on EM images. This model however does not work well on our datasets. We expect this is because microSAM has been finetuned on high-resolution EM images and does not generalize to our low-resolution datasets. We further compare TriSAM with the supervised method 3D U-Net (The MICrONS Consortium et al., 2023). We use the implementation from Wolny et al. (2020). Individual models are trained on $5\%$ of each dataset. We use $64 \times 192 \times 192$ dimensional cubes for training and train for 20000 iterations.

**Results Analysis.** The experimental results are shown in Table 2. First, both the Color Thresholding and SAM + IoU Tracking methods exhibit significant performance variability across various sub-datasets, highlighting the diversity of our dataset and the sensitivity of these methods to different species. Furthermore, both of these unsupervised methods demonstrate relatively poor performance, underscoring the challenges of the zero-shot setting in the BvEM dataset. Additionally, the 3D UNet, as a supervised learning approach, also yields very subpar results, indicating poor generalization of models trained with limited data. Finally, TriSAM outperforms the comparison methods by a large margin as it not only accurately segments the boundary but also tracks the blood vessels in the long term. Note that the results from the original dataset paper are also reported for reference.

Table 3: Ablation study results on different plan selection strategies. The proposed Tri-plane approach achieves the best overall accuracy with comparable speed.

| Method | Precision (%) | Recall (%) | Accuracy (%) | Speed (sec) |
|---|---|---|---|---|
| Single-Plane (z) | 75.28 | 48.48 | 41.82 | 324 |
| Single-Plane (y) | **79.37** | 58.47 | 50.75 | **307** |
| Single-Plane (x) | 69.73 | 12.14 | 11.53 | 345 |
| Single-Plane (fusion) | 71.78 | 74.15 | 57.41 | 976 |
| **Tri-Plane** | 78.41 | **74.97** | **62.14** | 335 |

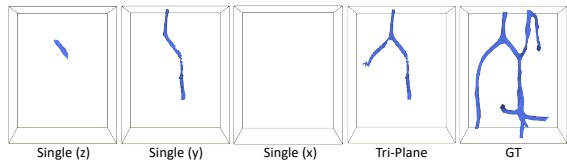

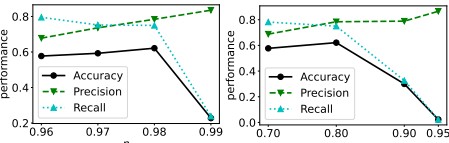

Figure 6: Comparison of segmentation results from one seed using different plane selection strategies.

Figure 7: Ablation study results on the hyperparameters $\eta$ and $\tau$.

**Qualitative Results.** The final instance segmentation results on BvEM-Macaque are shown in Figure 5. Color thresholding segments bright pixels, inadvertently capturing nuclei cells while overlooking darker pixels corresponding to blood vessels. Training the 3D UNet model with limited data results in confusion with background elements. IoU tracking fails to capture a significant portion of the blood vessel, revealing its ineffectiveness in tracking. TriSAM prediction emerges as the most accurate method, affirming the effectiveness of the proposed approach.

### 4.3 ABLATION STUDY

We conducted a comprehensive series of ablation studies exclusively using the BvEM-Macaque dataset due to the computation constraints.

**Effect of the Tri-Plane Selection.** We first compare our method with the single-plane methods to evaluate the effectiveness of Tri-Plane. For the single-plane method, we choose one plane as the main plane and only track along the chosen plane. The results on the BvEM-Macaque dataset are shown in Tabel 3. We have observed significant differences in performance among the three single-plane methods, with accuracy ranging from 11.53% to 50.75%, depending on the chosen tracking plane, which indicates the importance of the chosen tracking plane. This variability can be attributed to the tubular nature of blood vessel extensions within biological organisms, resulting in the generation of intricate mask shapes in certain planes, while simpler mask shapes are produced in others. Then we fuse the results of three Single-Plane methods and attain a higher accuracy of 57.41%, which demonstrates that the segmentation results from different planes exhibit a high degree of complementarity with each other. Instead, Tri-Plane exploits the blood vessel 3D structures by tracking the blood vessels along a suitable plane and attains the highest accuracy of 62.14%.

We visualize the segmentation results with one initial seed on BvEM-Macaque in Figure 6. We see that the performance is sensitive to the selection of the tracking plane. If the plane is not well-selected, the segmentation result can be empty as the example of tracking along the x-axis shows. The best result is tracking along the y-axis. However, it still falls short of the proposed method's performance, as the latter takes into account potential turning points to exploit the 3D structure and perform long-term tracking.

**Effect of the Recursive Redirection Strategy.** To perform long-term tracking and fully leverage the 3D blood vessel structure, we introduced recursive redirection by considering potential turning points. To validate its effectiveness, we report the results on the BvEM-Macaque dataset in Table 4, where the runtime comparison for segmentation prediction on the entire BvEM-Macaque data is also included. One naive baseline is to remove the recursive redirection component and not consider any potential turning points. This strategy is simple and fast but it fails to exploit the 3D shape prior leading to poor performance. Another strategy is to select the best plane for every tracking step/slice, which densely performs SAM segmentation on each step across three planes. Unfortunately, it significantly increases the computation cost. However, we were surprised to observe that

Table 4: Ablation study results on different redirection strategies. The proposed recursive redirection approach achieves the best overall accuracy with comparable speed.

| Strategy | Precision (%) | Recall (%) | Accuracy (%) | Speed (sec) |
|---|---|---|---|---|
| Naive | 79.37 | 58.47 | 50.75 | **307** |
| Dense Redirection | **86.94** | 24.84 | 23.95 | 2001(↑ 552%) |
| **Recursive Redirection** | 78.41 | **74.97** | **62.14** | 335 (↑ 9%) |

Table 5: Ablation study results on the choice of SAM models. The MobileSAM model achieves better performance with faster inference speed and smaller model size.

| Method | Pre (%) | Rec (%) | Acc (%) | Speed | Parameters |
|---|---|---|---|---|---|
| Ours + SAM (Kirillov et al., 2023) | 77.65 | 66.03 | 55.48 | 1535s | 615M |
| Ours + MobileSAM (Zhang et al., 2023) | **78.41** | **74.97** | **62.14** | **335s** | **9.66M** |

the performance of the Dense Redirection strategy was even worse. This could be attributed to the frequent axis changes potentially leading to the omission of certain parts of the blood vessel and causing splitting errors.

**Effect of the Mobile-SAM Model.** To further improve the efficiency, we use MobileSAM instead of the original SAM in practice. As shown in Table 5, the inference time of MobileSAM is 22% of the original SAM which confirms that MobileSAM significantly improves the inference speed. Moreover, the performance of MobileSAM is even better than the original SAM, possibly because the distilled small model is less prone to overfitting to the original natural image domain.

**Quality Analysis** We delve deeper into the plane dynamics in Figures 8 and 9. The majority of the selected plane predominantly tracks along the y-axis. This observation aligns with the experimental results presented in Table 3, where it is evident that a single plane tracking with the y-axis outperforms the z-axis and x-axis. This is because the blood vessel flows mainly along the y-axis as shown in Figure 6. In Figure 10, we explore the size dynamics using various methods. The size variation observed with our proposed method is relatively smaller compared to tracking along the y-axis and significantly less than when tracking along the z-axis and x-axis.

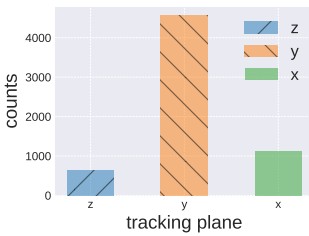

Figure 8: Histogram of tracking planes.

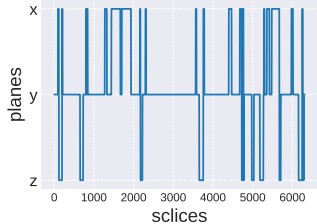

Figure 9: Dynamics of tracking planes.

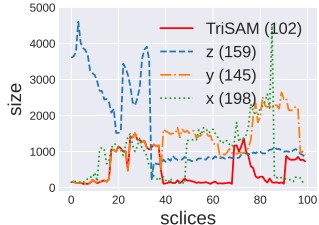

Figure 10: Dynamics of size during tracking. (X) indicates the mean derivation is X.

## 5 RELATED WORK

**Blood Vessel Segmentation.** Most existing VEM image segmentation algorithms were developed for neurons (Lee et al., 2019) and synapses (Wu et al., 2023a; Turner et al., 2020; Buhmann et al., 2021). The traditional VEM dataset size is too small to study blood vessel architecture. Recently, with rapid technology improvement, VEM sample size reached the cubic millimeter scale covering all the layers of cerebral cortex (Shapson-Coe et al., 2021; The MICrONS Consortium et al., 2023). This marks a significant milestone in connectomics research. The blood vessels were segmented manually in the human cortex (Shapson-Coe et al., 2021) and automatically using fully convolutional neural network (FCN) Tetteh et al. (2020) or 3D U-Net (The MICrONS Consortium et al., 2023; Livne et al., 2019; Çiçek et al., 2016). Due to the burden of annotation, efforts have been made to decrease the need for annotations (Dang et al., 2022). For images acquired using other related imaging methods, such as light microscopy and MRI, a variety of methods were developed and covered by some reviews (Goni et al., 2022; Moccia et al., 2018). Briefly, the blood vessels could be

enhanced based on the nature of tubeness using filters, such as diffusion filtering and ridge filtering. However, the segmentation of cortical blood vessels in VEM images is seldom explored, mainly because there is a lack of publicly available benchmarks.

**Evaluation Metrics.** Quantitative evaluation is normally conducted by comparing with manual annotations that are used as the ground truth. The evaluation metrics could be classified into two categories: pixel-based and skeleton-based (Moccia et al., 2018). Pixel-based metrics measure the difference between segmentation and ground truth at the pixel or voxel level, such as Accuracy and Dice Similarity Coefficient (Dice, 1945). Skeleton-based metrics measure the difference of skeletons or centerlines, such as Overlap and Overlap Until First Error (Metz et al., 2008). Skeleton-based metric requires the blood vessel skeleton that might not always be available. It also requires the computation of point-to-point correspondence between the ground truth and computed skeleltons. All existing evaluation metrics are developed using datasets acquired using other imaging methods and we did not find any specific metric for VEM datasets. Here we adopt the metrics Precision, Recall, and Accuracy as we are measuring instance-level pixel prediction.

**Segment Anything-Based Models.** As a foundation model for image segmentation, the recently proposed Segment Anything Model (Kirillov et al., 2023) has garnered significant attention. Due to its exceptional zero-shot performance, SAM has been extended to a variety of domains including object tracking (Cheng et al., 2023b; Yang et al., 2023), image inpainting (Yu et al., 2023), image mattting (Yao et al., 2023), super-resolution (Lu et al., 2023), 3D point cloud (Liu et al., 2023), and image editing (Gao et al., 2023). Despite SAM's remarkable generalization capabilities, it still encounters several challenges in practical applications. One of these challenges is the huge computation costs due to the heavyweight image encoder. FastSAM (Zhao et al., 2023) adopted a conventional CNN detector with an instance segmentation branch for the segment anything task with real-time speed. MobileSAM (Zhang et al., 2023) proposed decoupled distillation to obtain a small image encoder, which achieved approximately five times faster speed compared to FastSAM while also being seven times smaller in size. Therefore the MobileSAM is employed in our proposed method. Another challenge lies in the unsatisfactory performance of SAM when confronted with special domains, such as medical (Chen et al., 2023) or biological (Archit et al., 2023a) images, particularly in the context of 3D data. Deng et al. (2023) assessed the SAM model's zero-shot segmentation performance in the context of digital pathology and showed scenarios where SAM encounters difficulties. Mazurowski et al. (2023) extensively evaluates the SAM for medical image segmentation across 19 diverse datasets, highlighting SAM's performance variability based on prompts and dataset characteristics. To address the domain gap between natural and medical images, SAM-Adapter (Chen et al., 2023), SAM-Med2D (Cheng et al., 2023a), and Medical SAM Adapter (Wu et al., 2023b) introduced Adapter modules and trained the Adapter with medical images. They attained good performance on various medical image segmentation tasks. MedSAM (Ma & Wang, 2023) adapted SAM with more than one million medical image-mask pairs and attained accurate segmentation results. MicroSAM (Archit et al., 2023a) also presented a segment anything model for microscopy by fine-tuning SAM with microscopy data. Differing from these approaches that require model fine-tuning for adaptation, our method lifts the blood vessel segmentation capabilities of SAM from 2D images to 3D volumes without any model fine-tuning.

# 6 CONCLUSION

In this paper, we have contributed the largest-to-date public benchmark, the BvEM dataset, for cortical blood vessel segmentation in 3D VEM images. By addressing a significant gap in the field of neuroimaging, we have laid the foundation for advancing the understanding of cerebral vasculature at the microscale and its intricate relationship with neural function. We also developed a zero-shot blood vessel segmentation method, TriSAM, based on the powerful SAM model, offering an efficient and accurate approach for segmenting blood vessels in VEM images. With Tri-Plane selection, SAM-based tracking, and recursive redirection, our TriSAM effectively exploits the 3D blood vessel structure and attains superior performance compared with existing zero-shot and supervised technologies on BvEM across three species, marking a critical step towards unlocking the mysteries of neurovascular coupling and its implications for brain health and pathology. With the availability of the BvEM dataset and the TriSAM method, researchers are now equipped with valuable tools to drive breakthroughs in VEM-based cortical blood vessel segmentation and further our understanding of the brain's intricate vascular network.

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

## A    APPENDIX

### A.1    FAILURE CASES

Despite our method's significant progress, it still encounters challenges in some scenarios, as illustrated in Figure 11. The first issue is the seed problem, where the presence of a false positive seed can lead to erroneous predictions. Secondly, SAM could also fail to segment the mask. Lastly, tracking failure tends to occur at blood vessel conjunction points. These instances highlight the complexity of tracking in such intricate vascular networks and suggest the need for further refinements in our approach to address these specific failure cases effectively.

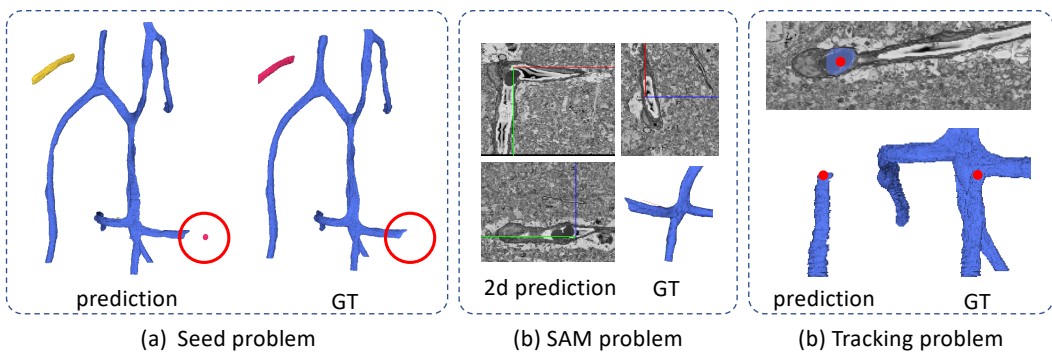

Figure 11: The failure cases of the proposed method. Three prediction and ground truth pairs from BvEM-Macaque are shown. Results are generated from one initial seed located at the origin point.

### A.2    ALGORITHM

The algorithm of TriSAM is shown in Algorithm 1.

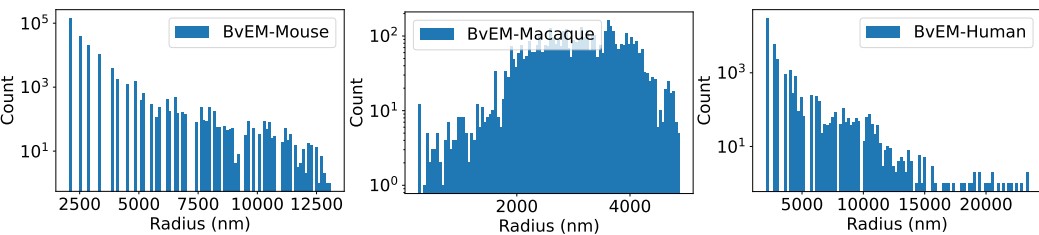

Figure 12: Dataset statistics: the histogram of the blood vessel radius.

---

**Algorithm 1** TriSAM algorithm

---

**Require:** 3D volume $\mathbf{X}$ and a sample seed s, a threshold $\tau$
  Initialize the prediction $\mathbf{P} = \mathbf{0}$
  Initialize the turning point tree $TurningPointTree = s$
  **for** $\mathrm{t}_i \quad in \quad$ UnsampledLeafNode($TurningPointTree$) **do**
    $plane = \mathrm{Select}(\mathbf{X}, \mathrm{t}_i)$                     ▷ Select the best tracking plane
    $segment = \mathrm{SAMBasedTrack}(\mathbf{X}, \mathrm{t}_i, plane)$      ▷ SAM-based tracking in the best plane
    $\mathbf{P} = \mathrm{Update}(\mathbf{P}, segment)$                ▷ Add the segment to $\mathbf{P}$
    $TurningPoints = \mathrm{RecursiveRedirect}(\mathbf{X}, \mathrm{t}_i)$      ▷ Generate potential turning points
    **if** $\mathrm{Valid}(TurningPoints)$ **then**
        $TurningPointTree = \mathrm{LeafSplit}(TurningPointTree, \mathrm{t}_i, TurningPoints)$    ▷ Grow
the turning point tree
    **end if**
  **end for**
  **return** $\mathbf{P}$

---

## A.3 DATASET STATISTICS

The histogram of the blood vessels in the proposed dataset is shown in Figure 12.

