# OpenReview forum: "TriSAM: Tri-Plane SAM for zero-shot cortical blood vessel segmentation in VEM images"
_ICLR.cc/2024/Conference — Submitted to ICLR 2024_

### Official Review · Reviewer_Qnji · 2023-10-29

**Soundness:** 2 fair
**Presentation:** 2 fair
**Contribution:** 3 good
**Rating:** 5
**Confidence:** 5

**Summary:**

This work introduces the TriSAM method, which is a zero-shot 3D segmentation method named TriSAM that relies on the Segment Anything Model (well-known as SAM). The framework can segment objects in an image given a point or bounding box as input. The designed framework is designed to segment blood vessels, hence the work proposes to integrate a multi-seed training strategy.

**Strengths:**

- Significance: The paper focuses on a very relevant problem that, to date, still remains unsolved.
- originality: The idea of combining a tracking approach with SAM sounds novel.

**Weaknesses:**

Clarity: The paper misses to provide precise details about how the method works. While the overall outline of the steps within TriSAM are very clear, how each of them are designed and formalized is not well explained in the paper.
Quality: There are aspects of the paper (see questions) that are not well justified. The experimental results do not consider state of the art methods on vessel segmentation (e.g. [1-3] to illustrate just a few examples), including some that reduce the annotation effort (see [2]).


[1] Livne, Michelle, et al. "A U-Net deep learning framework for high-performance vessel segmentation in patients with cerebrovascular disease." Frontiers in neuroscience 13 (2019): 97.
[2] Dang, Vien Ngoc, et al. "Vessel-CAPTCHA: an efficient learning framework for vessel annotation and segmentation." Medical Image Analysis 75 (2022): 102263
[3] Tetteh, Giles, et al. "Deepvesselnet: Vessel segmentation, centerline prediction, and bifurcation detection in 3-d angiographic volumes." Frontiers in Neuroscience 14 (2020): 1285.

**Questions:**

- What do the authors mean by this sentence " Moreover, imaging the whole mouse brain using VEM technology is under planning"?
- How are turning points detected?
- How is the tracking approach integrated with SAM?
- How is the model trained? The zero-shot aspect does not come across clear
- The paper states that : "By choosing the best plane during tracking, the shape and size will not change dramatically". This seems like a flawed argument. Across neighboring slides, the vessels should not dramatically change of size but progressively. However, as the brain vessels are tortuous, the change of shape can always occur.

---

> ### Comment · Reviewer_Qnji · 2023-11-21
>
> The authors have not submitted a rebuttal addressing the concerns raised during the first round of reviews. Hence, I will maintain my original score.

---

> ### Author Response · Authors · 2023-11-22
> **Response to Reviewer Qnji**
>
> Thanks for the insightful feedback. We clarify the issues and address the questions accordingly as described below.
>
> **Response Highlight**
>
> We would like to point out that one of our **main contributions is our constructed BvEM dataset**, which is the largest-to-date public benchmark for cortical blood vessel segmentation in VEM images. It addresses a significant gap in the neuroimaging field.
>
> **Q1: Experiment**
>
> **[Reply]**  We have conducted more experiments, such as nnUNET and MAESTER. Please refer to the answer to Q1 of General Response. We have cited and discussed the provided references [1-3] in Section 5.
>
> We also would like to highlight that the main focus of this paper is **zero-shot** vessel segmentation. As annotating 3D volumes for blood vessels is not only labor-intensive but also requires well-trained domain experts, efficient deep-learning algorithms are urgently needed. End-to-end training methods require annotating data for training in any new scenario. We deeply believe that in the era of foundational models, the zero-shot setting is the correct path to solving this problem. The emergence of the SAM model and the success of our proposed TriSAM have demonstrated this point. Comparing our approach to supervised learning methods is meaningless because supervised learning methods fall far short of the generalization capacity of our approach, and the annotation-training paradigm of supervised learning is far from what we desire. Therefore, the proposed BvEM dataset is also designed for evaluating zero-shot model performances, not only for training deep learning models. We reported the performance of 3D UNet solely for the completeness of the paper.
>
>
> **Q2: The meaning of "Moreover, imaging the whole mouse brain using VEM technology is under planning".**
>
> **[Reply]**  In response to the BRAIN Initiative, NIH has proposed a series of transformative projects. A major goal is scaling up the VEM technology to the whole mouse brain. A group of leading scientists calls for mapping a whole mouse brain in the next ten years. In response, a project mapping the connectome in a whole hippocampus has been granted as the first step. Considering the deep neuroscience background that is not familiar to computer scientists, we have deleted this sentence to avoid confusion.
>
>
> **Q3: Turning point detection**
>
> **[Reply]**  When tracking along the z-axis, we apply SAM to segment blood vessels in the zx and zy planes. Potential turning points are identified as points in these segments with the smallest and largest z-values. The process is detailed at the end of Section 3.
>
>
> **Q4: SAM-based tracking**
>
> **[Reply]**  SAM-based tracking: For the next slice, we use prompts derived from the enlarged bounding box and the segment's center in the current slice. Specifically, we pad the bounding box of the segment to create the bounding box prompt and calculate the segment's center for the point prompt. These prompts are used to segment blood vessels in the subsequent slice, as detailed in Section 3.2 "SAM-based Tracking."
>
> **Q5: Model training**
>
> **[Reply]**  Model training: Our proposed zero-shot method **eliminates** the need for model training. We utilize a pre-trained SAM for segmentation, leveraging its ability to segment objects from a point or bounding box. The initial points for segmentation are generated using an unsupervised color thresholding method. While this thresholding may miss significant portions of the blood vessel, these areas are effectively detected and segmented through SAM-based tracking.
>
>
> **Q6: Dynamics**
>
> **[Reply]** Blood vessels gradually change their shape and size in 3D, but the variations can significantly differ across different 2D planes, as illustrated in Figure 10 in the revised paper.
> We aim to highlight that along the optimally selected plane, changes in shape and size of the blood vessel are more stable **compared to other planes**. This statement has been updated in the revised version. We also compare the scale variation for different methods. Please refer to the answer to Q2 of the General Response.
>
> We hope our response clarifies your initial concerns/questions. We would be happy to provide further clarifications where necessary.

---

> ### Author Response · Authors · 2023-11-22
> **Response to Late Rebuttal**
>
> We apologize for the delay in submission, as fulfilling the reviewers' request for additional experiments on our proposed large-scale dataset proved to be a time-consuming process. Nonetheless, we managed to submit our response before the rebuttal deadline. As we have clarified some misunderstandings and addressed all your questions about our paper, we sincerely hope that you can reconsider the rating. Please feel free to let us know if you have any further questions.

---

### Official Review · Reviewer_BJ42 · 2023-11-01

**Soundness:** 3 good
**Presentation:** 2 fair
**Contribution:** 3 good
**Rating:** 5
**Confidence:** 4

**Summary:**

This paper presents the BvEM benchmark that provides Volume Electron Microscopy (VEM) image volumes from adult mice, macaque, and humans. Also, this proposes a zero-shot cortical blood vessel segmentation method, called TriSAM, which consists of Tri-Plane selection, SAM-based tracking, and recursive redirection. By choosing the best plane for tacking, this method enables effective long-term 3D blood vessel segmentation. The proposed method is demonstrated on the BvEM benchmark and shows superiority over the comparative methods.

**Strengths:**

- The paper proposes a new dataset, BvEM, which contains VEM images and their blood vessel segmentation labels verified by the experts.
- The proposed method extends the work of the Segment Anything Model (SAM) to 3D vessel segmentation, which addresses the problem of requiring large amounts of annotated training data.
- For SAM-based tracking, the authors select seeds in which the shape and size do not have dynamic changes by looking into three different planes.
- The proposed method is verified on the proposed benchmark dataset and achieves higher performance than the comparative methods.

**Weaknesses:**

- In the proposed method, the initial seed generation and triplane selection seem to be quite heuristic in that the selections depend on the threshold.
- The dynamics of the planes are not investigated in detail. The shape and size may be different along the images.
- There are many learning-based blood vessel segmentation methods, but only 3D UNet is used as a comparative method. The other Color Thresholding and SAM+IoU Tracking methods are not deep learning-based methods.
- There is a lack of description of why the proposed method adopts the SAM approach.

**Questions:**

- How the threshold for each initial seed selection and plane selection is determined? It seems difficult to find the optical threshold manually. Also, how are segmentation results different according to the threshold?
- Please discuss about the dynamic changes along the images for tracking blood vessels.
- Is the proposed TriSAM SOTA even when compared to the supervised image segmentation methods such as nnUNET?

---

> ### Author Response · Authors · 2023-11-22
> **Response to Reviewer BJ42**
>
> Thanks for the constructive comments. We answer your questions below:
>
> **Q1: Threshold**
>
> **[Reply]** The ablation study on the threshold for initial seed selection and triplane selection is detailed in Section 4.1, with results in Figure 7. We observed consistent method performance across various thresholds.
>
> As shown in Section 4.1,  the performance of the proposed method is largely insensitive to threshold variations, except at extremely high values, indicating our method's robustness to threshold settings.
>
>
> **Q2: Tracking Dynamics**
>
> **[Reply]** Please refer to the answer to Q2 of General Response.
>
> **Q3: Experiments**
>
> **[Reply]** We have conducted more experiments, such as nnUNET and MAESTER. Please refer to the answer to Q1 of General Response.
>
> We also would like to highlight that the main focus of this paper is **zero-shot** vessel segmentation. As annotating 3D volumes for blood vessels is not only labor-intensive but also requires well-trained domain experts, efficient deep-learning algorithms are urgently needed. End-to-end training methods require annotating data for training in any new scenario. We deeply believe that in the era of foundational models, the zero-shot setting is the correct path to solving this problem. The emergence of the SAM model and the success of our proposed TriSAM have demonstrated this point. Comparing our approach to supervised learning methods is meaningless because supervised learning methods fall far short of the generalization capacity of our approach, and the annotation-training paradigm of supervised learning is far from what we desire. Therefore, the proposed BvEM dataset is also designed for evaluating zero-shot model performances, not only for training deep learning models. We reported the performance of 3D UNet solely for the completeness of the paper.
>
>
> **Q4: Why SAM**
>
> **[Reply]** We chose SAM for its efficacy in segmenting objects with varied prompts like a point, box, or mask, and its excellent generalization to unseen data. In the era of foundational models, we firmly believe that for various segmentation tasks, a foundational model is all we need, and adaptations specific to different downstream tasks can be made based on this foundational model. As a foundational model for image segmentation, SAM has already been widely demonstrated for its strong generalization capabilities. More details have been included in the revised manuscript.
>
> We hope our response clarifies your initial concerns/questions. We would be happy to provide further clarifications where necessary.

---

### Official Review · Reviewer_NUw2 · 2023-11-01

**Soundness:** 3 good
**Presentation:** 3 good
**Contribution:** 3 good
**Rating:** 5
**Confidence:** 3

**Summary:**

The proposed TriSAM is based on the multi-seed tracking framework, which leverage specific image planes for tracking, while employing others to detect possible turning points. This framework is a combination of Tri-Plane selection, SAM-driven tracking, and recursive redirection. Evaluated on the proposed BvEM dataset, the proposed TriSAM is able to achieve long-term 3D blood vessel segmentation without the need for model training or fine-tuning.

**Strengths:**

1. A new benchmark of BvEM is introduced for blood vessel segmentation in volume electron microscopy images.
2. The proposed TriSAM works effectively and is able to achieve zero-shot 3D blood vessel segmentation.
3. The paper is well written and clearly organized.

**Weaknesses:**

1. The proposed method is more like an engineering implementation than a scientific research. It consists of four steps: Initial seed generation, Tri-plane selection, SAM-based tracking, and recursive redirection. Even though the rationale is simple, it works effectively.

2. The lack of comparison with sota VEM segmentation methods. The discussion of existing VEM segmentation methods are quited limited, more discussion should be provided to facilitate the understanding of existing researches. The proposed method might be compared with more SOTA zero-shot segmentation methods.

3. How to determine the threshold in Tri-Plane selection and SAM-based tracking? Do we need to change the value of threshold when applied to other data sets?

**Questions:**

1.  The existing VEM segmentation methods might be discussed in detail. What is the difference between the proposed method and existing works?

2. The threshold plays a vital role in the SAM-based tracking and recursive redirection. An ablation study of threshold could be provided to determine the influence of different values of threshold.

---

> ### Author Response · Authors · 2023-11-22
> **Response to Reviewer NUw2**
>
> Thanks for the constructive comments. We address the questions and clarify the issues accordingly as described below.
>
> **Q1: Novelty**
>
> **[Reply]** ***(a) Zero-shot approach***: One major challenge in 3D blood vessel segmentation is the limited labeled data to train effective deep learning models, as labeling 3D volumes is extremely time-consuming. Although existing unsupervised learning approaches eliminate the need for manual labeling, their 3D segmentation performance is not adequate. In contrast, we convert the 3D segmentation task into the tracking task and propose to integrate the existing foundation model, SAM, into our zero-shot pipeline achieving SOTA performance without any manual annotation.
> ***(b) SAM Adaptation***: Directly applying SAM to track blood vessel segments over slices does not achieve optimal performance (Table 2). Popular SAM-adaption methods need labeled data for finetuning. In contrast, the proposed SAM-based tracking pipeline is both effective and data efficient.
>
> **Q2: Experiment**
>
> **[Reply]** We further compare with nnUNET and MAESTER, please refer to the answer to Q1 of General Response.
>
> **Q3: Threshold**
>
> **[Reply]** The ablation study concerning the threshold is presented in Figure 7 and detailed in Section 4.1. We applied consistent hyperparameters across all species, which demonstrated effective generalization.
>
> **Q4: Discussion of Existing VEM Segmentation Methods**
>
> **[Reply]** Most existing VEM segmentation methods are supervised and need extensive labeled data, often limiting generalization across datasets. We summarized the existing VEM segmentation works in Section 5. Our method, requiring no labeled data, shows superior generalization to different datasets.
>
> We hope our response clarifies your initial concerns/questions. We would be happy to provide further clarifications where necessary.

---

### Author Response · Authors · 2023-11-22
**General Response**

We would like to thank the reviewers for their thoughtful comments.  We sincerely thank the reviewers for recognizing our **"new dataset"** (Reviewer NUw2 and BJ42), considering our approach to **"work effectively"** (Reviewer NUw2 and BJ42), and finding the idea of our method to be **"novel"** (Reviewer Qnji).
We carefully addressed each issue and revised the manuscript accordingly.

**Q1: More Quantitative Baseline Comparisons [For Reviewers NUw2, BJ42, and Qnji]**

**[Reply]**  Upon reviewers' suggestions, we added comparisons to include the state-of-the-art unsupervised method (MAESTER), and another state-of-the-art supervised method (nnUNET). The additional results demonstrate that our proposed TriSAM method outperforms both methods. This table corresponds to Table 2 in our revised paper.

**Table 2. Benchmark results on the proposed BvEM dataset.**

| Method             | Training Data   |   BvEM-Mouse Pre |   BvEM-Mouse Rec |   BvEM-Mouse Acc |   BvEM-Macaque Pre |   BvEM-Macaque Rec |   BvEM-Macaque Acc |   BvEM-Human Pre |   BvEM-Human Rec |   BvEM-Human Acc |
|:-------------------|:----------------|-----------------:|-----------------:|-----------------:|-------------------:|-------------------:|-------------------:|-----------------:|-----------------:|-----------------:|
| Initial Annotation | Unknown         |            93.74 |            36.62 |            35.74 |               1.65 |              23.17 |               1.57 |           100    |            25.68 |            25.68 |
| 3D UNet            | Limited         |            13.45 |             0.91 |             0.67 |              16.04 |              89.65 |              15.75 |            68.63 |             2.46 |             2.43 |
| nnUNET             | Limited         |             3.54 |            49.55 |             6.59 |              24.34 |              29.57 |              23.74 |             3.44 |            23.2  |             5.86 |
| Color Thresholding | None            |            86.45 |            37.32 |            35.26 |              95.14 |              21.65 |              21.42 |            41.77 |             1.92 |             1.87 |
| MAESTER            | None            |             2.16 |            18.95 |             0.94 |              22.08 |              40.3  |              16.64 |             0.29 |             5.03 |             0.27 |
| SAM + IoU Tracking | External        |            63.59 |             0.27 |             0.27 |              74.39 |               1.89 |               1.88 |            18.19 |            23.58 |            12.92 |
| TriSAM             | External        |            84.12 |            66.75 |            59.28 |              78.41 |              74.97 |              62.14 |            31.35 |            25.57 |            16.39 |



**Q2: More Qualitative Analysis on the Tracking Dynamics [For Reviewers BJ42 and Qnji]**

**[Reply]**  We have added three figures (Figures 8, 9, and 10) in our revised paper to visualize the process. Specifically, we delve deeper into the plane dynamics in Figures 8 and 9. The majority of the selected plane predominantly tracks along the y-axis. This observation aligns with the experimental results presented in Table 3, where it is evident that a single plane tracking with the y-axis outperforms those tracking with the z-axis and x-axis. This is because the blood vessel flows mainly along the y-axis as shown in Figure 6. In Figure 10, we explore the size dynamics using various methods. The size variation observed with our proposed method is relatively smaller compared to tracking along the y-axis and significantly less than when tracking along the z-axis and x-axis.

---

### Meta-Review · Area_Chair_UCe1 · 2023-12-06

**Metareview:**

This paper present a new dataset for vessel segmentation in volume electron microscopy images as well as a zero shot approach that combines SAM with multi-seeds tracking. Overall, the paper is easy to follow. A major weakness of the method is the heuristic parameter settings that are important to the performance. This actually raises a concern if this method is fully zero shot. In my opinion, the selection of the parameter itself could be considered as part of learning. It would be more convincible if the default parameters (based on experience in other applications, for example) are used and yield promising results. However, based on the authors' ablation study, the results are sensitive to the parameters, which means we may need to learn these parameters. The technical novelty of the approach to combine SAM with tracking algorithms is limited as well. The reviewers still have some concern on the experimental comparisons.

Another contribution of the work seems to be the dataset, but it is not clear if the authors would release the data & annotation for public use.  In addition, proper approval such as IRB shall be in placed and mentioned.

**Justification For Why Not Higher Score:**

The contribution of this work is limited. From technology point of view, the novelty is quite limited as it is a simple combination of two approaches.

**Justification For Why Not Lower Score:**

NA

---

### Decision · Program_Chairs · 2024-01-16

Reject